# REPRESENTATION-CONSTRAINED AUTOENCODERS AND AN APPLICATION TO WIRELESS POSITIONING

## ABSTRACT

In a number of practical applications that rely on dimensionality reduction, the dataset or measurement process provides valuable side information that can be incorporated when learning low-dimensional embeddings. We propose the inclusion of pairwise representation constraints into autoencoders (AEs) with the goal of promoting application-specific structure. We use synthetic results to show that only a small amount of AE representation constraints are required to substantially improve the local and global neighborhood preserving properties of the learned embeddings. To demonstrate the efficacy of our approach and to illustrate a practical application that naturally provides such representation constraints, we focus on wireless positioning using a recently proposed channel charting framework. We show that representation-constrained AEs recover the global geometry of the learned low-dimensional representations, which enables channel charting to perform approximate positioning without access to global navigation satellite systems or supervised learning methods that rely on extensive measurement campaigns.

## 1 INTRODUCTION

Autoencoders (AEs) are single- or multi-layer neural networks that are widely used for dimensionality-reduction tasks (Hinton & Salakhutdinov, 2006; van der Maaten et al., 2009; Goodfellow et al., 2016; Baldi, 2012). AEs learn low-dimensional representations (also known as embeddings) of a given high-dimensional dataset and have been shown to accurately preserve spatial relationships in both high- and low-dimensional space for a broad range of synthetic and real-world datasets (van der Maaten et al., 2009). With the success of deep neural networks, AEs are also gaining increased attention for unsupervised learning tasks (Geng et al., 2015). Notable application examples of AEs include: learning word embeddings (Mikolov et al., 2013), unsupervised pretraining of deep networks to improve robustness (Erhan et al., 2010), imputation of missing data (Abdella & Marwala, 2005; Qiu et al., 2018), image compression (Theis et al., 2017), extracting features that are robust to data corruption (Vincent et al., 2008), and generative models (Bengio et al., 2013). In all of these applications, AEs are typically trained in an unsupervised manner, i.e., no labels are used, potential side information on the training data is ignored, or application-specific representation structure is not imposed during training.

AEs that impose structural constraints on the latent variables include sparse AEs (Hinton et al., 2006; Makhzani & Frey, 2013) and variational AEs (Doersch, 2016). Sparse AEs enforce sparsity on the representations, which enables one to learn embeddings with low effective dimensionality caused by the sparsity constraints. Variational AEs are able to learn a latent variable (representation) drawn from a distribution that represents the high-dimensional input (Doersch, 2016); such AEs have been shown to be able to generate complex data, such as handwritten digits (Kingma & Welling, 2013), faces (Kulkarni et al., 2015), house numbers (Kingma et al., 2014), or physical scene models (Kulkarni et al., 2015).

### 1.1 AUTOENCODERS WITH REPRESENTATION CONSTRAINTS

Certain applications provide additional constraints or side information that can be imposed on the low-dimensional representations. Such side information may stem either from the dataset itself or from the application (e.g., from the way data was collected). One example arises when data is

acquired over time. In such a scenario it may be natural to enforce constraints *between* representations by exploiting the fact that for temporally correlated datapoints, the associated low-dimensional representations should exhibit a certain degree of similarity. Another example arises in situations in which a subset of the representations are known a-priori, e.g., when a subset of training data is annotated. The obtained annotation information can then be translated into representation constraints, which leads to semi-supervised training of an AEs.

A concrete application example in which *representation constraints* are important is positioning of users in wireless communication systems using channel charting (CC) (Studer et al., 2018). CC measures high-dimensional channel state information (CSI) of users transmitting data to a wireless access point or cell tower. By collecting CSI over time, one can train an AE for which the low-dimensional representations reflect relative user positions. While the method in (Studer et al., 2018) enables logical (or relative) localization without access to global navigation satellite systems (GNSS) and without expensive measurement campaigns (Liu et al., 2007; Xu et al., 2016), valuable side information should not be ignored when available. For example, users can only move with finite velocity. One could include this information when training an AE to ensure that temporally correlated datapoints are nearby in the representation space. In addition, one could imagine that certain points in space with known location (e.g., a coffee shop) can be associated with measured CSI; this helps to pin down a small set of spatial locations in the representation space, which may enable absolute positioning in space. Put simply, by enforcing constraints *between* representations, one can hope to improve the efficacy of AEs and enable interpretability of the learned low-dimensional representations.

## 1.2 CONTRIBUTIONS

This paper investigates four distinct representation constraints for AEs and provides a framework for including these during training. We propose constraints on pairs of representations in which either the absolute or relative distance among (a subset of) representations is enforced. We formulate these constraints as nonconvex regularizers, which can easily be included into existing deep-learning frameworks. We provide synthetic experiments to demonstrate that only a small amount of representation constraints are required to (often significantly) improve the representation quality of AEs. As a byproduct of our framework, we show that one of the proposed constraint types enables one to learn a parametric version of Sammon's mapping (Sammon, 1969) that avoids the need of costly out-of-sample extensions. We highlight the efficacy of representation constraints for the application of CC-based positioning in wireless systems. In an application context, we demonstrate that by combining partially-annotated datapoints with temporal constraints that arise from the acquisition process, the positioning performance of channel charting (Studer et al., 2018) can be improved significantly.

## 1.3 RELEVANT PRIOR ART

Autoencoders have been shown to consistently outperform other dimensionality-reduction algorithms on real-world datasets (van der Maaten et al., 2009). Despite their success, AEs do not aim at preserving geometric properties on the representations (such as distances between points), which is in stark contrast to, e.g., Sammon's mapping (Sammon, 1969) or multidimensional scaling (Davison, 1991). Variations on classical AEs, such as denoising AEs, convolutional AEs, sparse AEs, or variational AEs, have been shown to further improve performance of the learned representations. van der Maaten (2009) proposes the use of AEs to learn a parametric mapping between high-dimensional datapoints and low-dimensional representations by enforcing structure obtained via Student-t stochastic neighborhood embedding (t-SNE). None of these methods incorporate potential side information that stems from the analyzed or the application at hand. We will extend AEs with *representation constraints* that further improve performance in certain applications. We note that our methods can be used in combination with existing techniques.

Metric learning deals with the design of suitable measures of similarity (Kulis et al., 2013; Xing et al., 2002; Weinberger & Saul, 2009; Baghshah & Shouraki, 2009; Hoi et al., 2010). The use of side information for metric learning has shown to (often significantly) improve performance (Xing et al., 2002; Kumar & Kummamuru, 2008). Common constraints enforce that "similar" points lie close together in latent space (Xing et al., 2002). The methods we propose resemble those of (Klein

et al., 2002; Kumar & Kummamuru, 2008; Friedman, 1994; Schultz & Joachims, 2004; Baghshah & Shouraki, 2009) with the key difference that we are not interested in learning a metric but rather a representation that preserves structure provided by the application and dataset.

Related to applications of interest, machine learning has slowly found its way into wireless communication systems. Past research has largely focused on replacing detection or decoding with deep neural networks (O'Shea & Hoydis, 2017). More recently, the efficacy of deep learning has been explored for wireless positioning (Chen et al., 2017; Vieira et al., 2017; Arnold et al., 2018). All of these methods require extensive measurement campaigns in which one has to annotate CSI measurements with exact position information obtained from GNSS, such as the global positioning system (GPS). To avoid the drawback of such supervised methods, channel charting (CC) (Studer et al., 2018) uses dimensionality reduction techniques to extract *relative* positions of users without the necessity of costly measurement campaigns. CC exploits the fact that CSI is high-dimensional, but it strongly depends on a user's position, which is low-dimensional. Dimensionality reduction (e.g., via Sammon's mapping or AEs) applied to CSI measurements can be used to learn a so-called *channel chart* in which nearby points represent nearby positions in true space—exact position information is not available. However, wireless positioning provides a unique set of representation constraints originating from the acquisition process, which have been ignored in prior work. We propose to include such constraints into AEs which enables significantly improved CC performance and paves the way for exact positioning via CC.

## 2 REPRESENTATION-CONSTRAINED AUTOENCODERS

We briefly summarize the basics of AEs, introduce representation constraints, and then show a range of synthetic results that demonstrate the efficacy of representation constraints.

### 2.1 AUTOENCODERS IN A NUTSHELL

AEs take a high-dimensional dataset consisting of $N$ datapoints (vectors) $\mathbf{x}_n \in \mathbb{R}^D$, $n = 1, \ldots, N$, of dimension $D$, and learn two functions: the encoder $f^e : \mathbb{R}^D \to \mathbb{R}^{D'}$ and the decoder $f^d : \mathbb{R}^D \to \mathbb{R}^{D'}$. The encoder maps datapoints onto low-dimensional *representations* $\mathbf{y}_n \in \mathbb{R}^{D'}$, $n = 1, \ldots, N$, where $D' \ll D$ is the dimension of the representations, and the decoder maps representations back to datapoints, i.e., we have

$$\mathbf{y}_n = f^e(\mathbf{x}_n) \quad \text{and} \quad \mathbf{x}_n = f^d(\mathbf{y}_n), \quad n = 1, \ldots, N.$$

The encoder and decoder functions of AEs are implemented as multilayer (shallow or deep) feed-forward neural networks that are trained to minimize the mean-square error (MSE) between the input and the output of the network. Specifically, one seeks to minimize the following loss function:

$$L(\mathcal{W}^e, \mathcal{W}^d) = \frac{1}{N} \sum_{n=1}^{N} \|\mathbf{x}_n - f^d(f^e(\mathbf{x}_n))\|^2. \tag{1}$$

Here, the parameters to be learned from the dataset $\{\mathbf{x}_n\}_{n=1}^{N}$ are the weights and bias terms contained in the sets $\mathcal{W}^e$ and $\mathcal{W}^d$ that define the neural network forming the encoder and decoder, respectively. All norms used in this paper are $\ell_2$-norms. While training of AEs is notoriously difficult, numerous strategies have been proposed in the past that improve the quality of training and reduce the probability of getting stuck in local minima; see, e.g., (van der Maaten et al., 2009; Kingma & Welling, 2013), for the details.

The $D'$-dimensional output of the encoder $f^e$ is typically of lower dimension than the intrinsic dimension of the manifold embedding the inputs $\mathbf{x}_i$ in $D$ dimensions. Hence, we have that $\mathbf{x}_n \approx f^d(f^e(\mathbf{x}_n))$, $n = 1, \ldots, N$, unless the dataset $\{\mathbf{x}_n\}_{n=1}^{N}$ was $D'$-dimensional and we were able to learn the underlying structure. Nevertheless, AEs often find low-dimensional representations $\{\mathbf{y}_n\}_{n=1}^{N}$ with small loss that capture the intrinsic dimensionality of the input datavectors.

### 2.2 REPRESENTATION CONSTRAINTS

We now propose four distinct representation constraints summarized in Table 1. In what follows, underlined quantities represent *constant* scalars or vectors that are known a-priori and used during AE training. Non-underlined quantities are optimization variables.

Table 1: Summary of proposed representation constraints for AEs (known quantities are underlined).

| Name | Constraint | Regularizer |
|------|-----------|-------------|
| Fixed absolute distance (FAD) | $\|\mathbf{y}_i - \underline{\mathbf{y}}_j\| = \underline{d}_{i,j}$ | $(\|\mathbf{y}_i - \underline{\mathbf{y}}_j\| - \underline{d}_{i,j})^2$ |
| Fixed relative distance (FRD) | $\|\mathbf{y}_i - \mathbf{y}_j\| = \underline{d}_{i,j}$ | $(\|\mathbf{y}_i - \mathbf{y}_j\| - \underline{d}_{i,j})^2$ |
| Maximum absolute distance (MAD) | $\|\mathbf{y}_i - \underline{\mathbf{y}}_j\| \leq \underline{d}_{i,j}$ | $\max\{\|\mathbf{y}_i - \underline{\mathbf{y}}_j\| - \underline{d}_{i,j}, 0\}^2$ |
| Maximum relative distance (MRD) | $\|\mathbf{y}_i - \mathbf{y}_j\| \leq \underline{d}_{i,j}$ | $\max\{\|\mathbf{y}_i - \mathbf{y}_j\| - \underline{d}_{i,j}, 0\}^2$ |

**Fixed Distance Constraints**  The fixed absolute distance (FAD) and fixed relative distance (FRD) constraints enforce a known distance $\underline{d}_{i,j}$ on a pair of representations according to $\|\mathbf{y}_i - \mathbf{y}_j\| = \underline{d}_{i,j}$. The difference between the FAD and FRD constraints is that for FAD one of the two representations, e.g., $\underline{\mathbf{y}}_j$, is a constant known prior to AE learning (i.e., not to be confused with a representation in the dataset); for FRD, both representations $\mathbf{y}_i$ and $\mathbf{y}_j$ are optimization variables. To ease the inclusion of these constraints in deep learning frameworks, we propose to use regularizers (see Table 1) for which generalized gradients exist. Concretely, the generalized gradient of the FAD and FRD constraints with respect to representation $\mathbf{y}_i$ is

$$\nabla_{\mathbf{y}_i}(\|\mathbf{y}_i - \mathbf{y}_j\| - d_{i,j})^2 = 2(\|\mathbf{y}_i - \mathbf{y}_j\| - \underline{d}_{i,j})\frac{\mathbf{y}_i - \mathbf{y}_j}{\|\mathbf{y}_i - \mathbf{y}_j\|}, \qquad (2)$$

where the representation $\underline{\mathbf{y}}_j$ is known for FAD. If $\underline{d}_{i,j} = 0$, then the FRD regularizer promotes the two representations $\mathbf{y}_i$ and $\mathbf{y}_j$ to be equal, whereas the FAD regularizers will try to learn a representation $\mathbf{y}_i$ that is close to the constant vector $\underline{\mathbf{y}}_j$. Intuitively, the FAD constraint for $\underline{d}_{i,j} = 0$ acts as a semi-supervised extension in which one knows parts of the representations a-priori.

It is interesting to see that the FRD regularizer resembles that of Sammon's mapping (Sammon, 1969). In fact, as we will show in Section 2.2.1, by including FRD regularizers on all datavectors one can train a parametric AE version of Sammon's mapping if we multiply a factor of $\underline{d}_{i,j}^{-1}$ to each FRD regularizer term.

**Maximum Distance Constraints**  The maximum absolute distance (MAD) and maximum relative distance (MRD) constraints enforce a maximum a-priori known distance $\underline{d}_{i,j}$ between a pair of representations according to $\|\mathbf{y}_i - \mathbf{y}_j\| \leq \underline{d}_{i,j}$. For MAD, one of the two vectors in the constraint, e.g., $\underline{\mathbf{y}}_j$, is constant that is known a-priori; for MRD, both representations are learned. We include these constraints as regularizers (see Table 1) with the generalized gradient

$$\nabla_{\mathbf{y}_i} \max\{\|\mathbf{y}_i - \mathbf{y}_j\| - d_{i,j}, 0\}^2 = 2\max\{\|\mathbf{y}_i - \mathbf{y}_j\| - d_{i,j}, 0\}\frac{\mathbf{y}_i - \mathbf{y}_j}{\|\mathbf{y}_i - \mathbf{y}_j\|}, \qquad (3)$$

where $\underline{\mathbf{y}}_j$ is known for MAD. As we show in Section 2.2.1, maximum distance regularizers often yield superior results to their fixed-distance counterparts as they leave the AE more "freedom" while learning representations. Note, when $d_{i,j} = 0$ the FAD, MAD, FRD, and MRD are all equivalent.

**Practical Considerations**  We implemented a stochastic optimizer to minimize the sum of the AE fidelity term (1) and the regularized constraint penalties using the Keras machine learning framework (Abadi et al., 2015; Chollet et al., 2015). Because penalty terms may represent pairwise constraints that involve two data points, the stochastic approximation of the regularizers was formed by randomly sampling constraints rather than data-points.

To improve numerical robustness of SGD for the generalized gradients in (2) and (3), we use the approximation $\frac{\mathbf{y}_i - \mathbf{y}_j}{\|\mathbf{y}_i - \mathbf{y}_j\|} \approx \frac{\mathbf{y}_i - \mathbf{y}_j}{\rho + \|\mathbf{y}_i - \mathbf{y}_j\|}$, where $\rho \geq 0$ is set to a small constant.

### 2.2.1 SYNTHETIC EXPERIMENTS

We now provide synthetic results for four standard datasets, namely the Swiss roll, the broken Swiss roll, the twin-peaks, and the helix datasets, and the four representation constraints in Table 1.

**Performance Metrics** We use two standard metrics for the local neighborhood-preserving performance of dimensionality reduction tasks: *trustworthiness* (TW) and *continuity* (CT). Concretely, the TW measures whether mapping high-dimensional datapoints to representation space introduces new (false) neighbors. TW is defined as follows:

$$TW(K) = 1 - \frac{2}{NK(2N - 3K - 1)} \sum_{i=1}^{N} \sum_{j \in \mathcal{U}_i^K} (r(i, j) - K).$$

Here, $r(i, j)$ represents the rank of the representation $\mathbf{y}_i$ among the pairwise distances between the other representations. The set $\mathcal{U}_i^K$ contains the points that are among the $K$ nearest neighbors in representation space but not in high-dimensional space. The CT measures whether similar datapoints in original space remain similar in representation space. CT is defined as follows:

$$CT(K) = 1 - \frac{2}{NK(2N - 3K - 1)} \sum_{i=1}^{N} \sum_{j \in \mathcal{V}_i^K} (\hat{r}(i, j) - K).$$

Here, $\hat{r}(i, j)$ represents the rank of the datapoint $\mathbf{x}_i$ among to the pairwise distances between the other datavectors. The set $\mathcal{V}_i^K$ contains the set of points that are among the $K$ nearest neighbors in high-dimensional space but not in representation space. For both the TW and CT, the values are in the range $[0, 1]$ and large values indicate better local-neighborhood-preserving properties.

In addition to measuring the local-neighborhood-preservation properties via TW and CT, we also consider Kruskal's stress (KS) (Lee & Verleysen, 2009; Shepard, 1962), which measures how well the global structure in the high-dimensional dataset $\{\mathbf{x}_n\}_{n=1}^{N}$ is mapped to the low-dimensional embedding $\{\mathbf{y}_n\}_{n=1}^{N}$. The KS is computed as follows:

$$KS = \sqrt{\frac{\sum_{n,m}(\delta_{n,m} - \beta\hat{\delta}_{n,m})^2}{\sum_{n,m} \delta_{n,m}^2}},$$

where $\delta_{n,m} = \|\mathbf{x}_n - \mathbf{x}_m\|$, $\hat{\delta}_{n,m} = \|\mathbf{y}_n - \mathbf{y}_m\|$, and $\beta = \sum_{n,m} \delta_{n,m}\hat{\delta}_{n,m} / \sum_{n,m} \delta_{n,m}^2$ is the optimal distance scaling factor. The KS is in the range $[0, 1]$ and smaller values indicate that global structure is preserved better. If $KS = 0$, then the structure is perfectly preserved.

**Datasets** In the following synthetic experiments, we use the same AE topology for all datasets and constraints. The AE consists of three hidden layers for the encoder and the decoder (each layer has 9, 7, and 3 neurons with rectified linear units). The encoder output that extracts the representation consists of 2 neurons with linear activation functions. The datasets are generated as described in van der Maaten et al. (2009). For each dataset, we generate $N = 5000$ points and we add i.i.d. zero-mean Gaussian noise with variance $\sigma^2 = 0.05$.

**Results** Table 2 shows the TW, CT, and KS for the FAD, FRD, MAD, and MRD representation constraints. We vary the fraction $\varepsilon$ of representations for which we impose constraints; $\varepsilon = 0$ means no constraints are imposed, $\varepsilon = 1$ means all datapoints have constraints. We evaluate the TW and CT for $K = 1$, for $K = 125$ (2.5% of the dataset) and $K = 250$ (5% of the dataset). For most datasets, the TW, CT, and KT improves by adding as little as 1% representation constraints. We observe that the relative constraints often outperform their absolute counterparts; we attribute this to the fact that relative constraints allow the AE more "freedom" to learn a representation. We furthermore see that fixed constraints generally outperform maximum constraints. One exception is the helix dataset for which we obtain mixed results.

As shown next, we can find excellent embeddings for this dataset when using a different approach to impose representation constraints. Noting that the datasets themselves provide tentative distances in the representation set, we incorporate these as Sammon's mapping constraints to the AEs. For this, we use FRD regularizers, scaled with corresponding factors of $\underline{d}_{i,j}^{-1}$, as discussed in Section 2.2. Figure 1 shows results of such Sammon's-enhanced AEs. We train the AEs for the same datasets used in Table 2 and provide TW and CT values for $K = 250$ (5% of the dataset). Clearly, AEs extended with the capabilities of Sammon's mapping are able to almost perfectly unfold the manifolds underlying each dataset, except for the broken Swiss roll. Note that the proposed representation constraints are

Table 2: TW, CT, and KS results for various representation constraints.

| | $\varepsilon$ | Swiss roll | | | Broken Swiss roll | | | Twin-peaks | | | Helix | | |
|---|---|---|---|---|---|---|---|---|---|---|---|---|---|
| | | 0% | 1% | 10% | 0% | 1% | 10% | 0% | 1% | 10% | 0% | 1% | 10% |
| | | Fixed absolute distance (FAD) | | | | | | | | | | | |
| TW | $K=1$ | 0.7984 | 0.7979 | **0.8654** | 0.8204 | **0.8486** | 0.8339 | 0.8746 | 0.9461 | **0.9742** | 0.9911 | 0.9759 | **0.9934** |
| | $K=125$ | 0.7063 | 0.7354 | **0.8226** | 0.7196 | **0.8083** | 0.7870 | 0.8879 | 0.9446 | **0.9721** | 0.9470 | 0.9647 | **0.9834** |
| | $K=250$ | 0.6584 | 0.7177 | **0.7991** | 0.6689 | **0.7940** | 0.7731 | 0.8965 | 0.9437 | **0.9698** | 0.9138 | 0.9526 | **0.9694** |
| CT | $K=1$ | 0.9950 | 0.9959 | **0.9981** | 0.9970 | 0.9987 | **0.9988** | 0.9671 | 0.9739 | **0.9897** | 0.9933 | 0.9940 | **0.9950** |
| | $K=125$ | 0.8553 | **0.9652** | 0.9567 | 0.8955 | **0.9778** | 0.9728 | 0.9289 | 0.9461 | **0.9795** | 0.9905 | 0.9943 | **0.9963** |
| | $K=250$ | 0.7855 | **0.9403** | 0.9249 | 0.8303 | **0.9623** | 0.9506 | 0.9097 | 0.9300 | **0.9760** | 0.9709 | 0.9819 | **0.9867** |
| KS | | **0.2359** | 0.2572 | 0.2452 | 0.1983 | **0.1863** | 0.1965 | 0.5105 | 0.1804 | **0.0853** | **0.2032** | 0.2354 | 0.2166 |
| | | Fixed relative distance (FRD) | | | | | | | | | | | |
| TW | $K=1$ | 0.7984 | **0.8896** | 0.9820 | 0.8204 | 0.8268 | **0.8284** | 0.8746 | 0.9244 | **0.9828** | **0.9911** | 0.9689 | 0.9852 |
| | $K=125$ | 0.7063 | 0.8365 | **0.9683** | 0.7196 | 0.7804 | **0.7887** | 0.8879 | 0.9151 | **0.9784** | 0.9470 | 0.9230 | **0.9778** |
| | $K=250$ | 0.6584 | 0.8184 | **0.9632** | 0.6689 | 0.7671 | **0.7719** | 0.8965 | 0.9088 | **0.9715** | 0.9138 | 0.9012 | **0.9718** |
| CT | $K=1$ | 0.9950 | **0.9981** | 0.9980 | **0.9970** | 0.9969 | 0.9740 | 0.9671 | 0.9827 | **0.9856** | **0.9933** | 0.9859 | 0.9928 |
| | $K=125$ | 0.8553 | 0.9678 | **0.9799** | 0.8955 | 0.9523 | **0.9552** | 0.9289 | 0.9671 | **0.9810** | 0.9905 | 0.9778 | **0.9940** |
| | $K=250$ | 0.7855 | 0.9478 | **0.9695** | 0.8303 | 0.9271 | **0.9402** | 0.9097 | **0.9558** | 0.9854 | 0.9709 | 0.9479 | **0.9863** |
| KS | | 0.2359 | **0.1894** | 0.2206 | **0.1983** | 0.2241 | 0.2173 | 0.5105 | **0.1437** | 0.2101 | 0.2032 | 0.3634 | **0.1911** |
| | | Maximum absolute distance (MAD) | | | | | | | | | | | |
| TW | $K=1$ | 0.7984 | 0.8744 | **0.8898** | 0.8204 | 0.8154 | **0.9099** | 0.8746 | 0.8120 | **0.9671** | **0.9911** | 0.9291 | 0.9596 |
| | $K=139$ | 0.7063 | 0.8261 | **0.8366** | 0.7196 | 0.7828 | **0.8675** | 0.8879 | 0.7884 | **0.9491** | **0.9470** | 0.8326 | 0.9087 |
| | $K=250$ | 0.6584 | 0.8137 | **0.8220** | 0.6689 | 0.7762 | **0.8576** | 0.8965 | 0.7787 | **0.9431** | **0.9138** | 0.8120 | 0.8847 |
| CT | $K=1$ | 0.9950 | **0.9968** | 0.9962 | 0.9970 | 0.9938 | **0.9977** | 0.9671 | 0.9057 | **0.9809** | 0.9933 | 0.9708 | **0.9888** |
| | $K=139$ | 0.8553 | 0.9290 | **0.9414** | 0.8955 | 0.9241 | **0.9493** | 0.9289 | 0.8500 | **0.9594** | 0.9905 | 0.9405 | **0.9814** |
| | $K=250$ | 0.7855 | 0.8946 | **0.9161** | 0.8303 | 0.8956 | **0.9216** | 0.9097 | 0.8227 | **0.9439** | 0.9709 | 0.8998 | **0.9593** |
| KS | | 0.2359 | 0.2899 | **0.2307** | **0.1983** | 0.2865 | 0.2442 | 0.5105 | 0.3158 | **0.2458** | **0.2032** | 0.3440 | 0.2981 |
| | | Maximum relative distance (MRD) | | | | | | | | | | | |
| TW | $K=1$ | 0.7984 | 0.8677 | **0.8835** | 0.8204 | 0.8056 | **0.9023** | 0.8746 | 0.9846 | **0.9599** | **0.9911** | 0.8458 | 0.9561 |
| | $K=139$ | 0.7063 | 0.8189 | **0.8278** | 0.7196 | 0.7750 | **0.8580** | 0.8879 | **0.9816** | 0.9526 | **0.9470** | 0.8061 | 0.9344 |
| | $K=250$ | 0.6584 | 0.8067 | **0.8128** | 0.6689 | 0.7660 | **0.8414** | 0.8965 | **0.98** | 0.9470 | 0.9138 | 0.8006 | **0.9293** |
| CT | $K=1$ | 0.9950 | **0.9970** | 0.9963 | **0.9970** | 0.9930 | 0.9884 | 0.9671 | 0.9660 | **0.9792** | **0.9933** | 0.9734 | 0.9879 |
| | $K=139$ | 0.8553 | 0.9276 | **0.9427** | 0.8955 | 0.9189 | **0.9645** | 0.9289 | **0.9666** | 0.9609 | 0.9905 | 0.9414 | **0.9826** |
| | $K=250$ | 0.7855 | 0.8917 | **0.9192** | 0.8303 | 0.8894 | **0.9440** | 0.9097 | **0.9673** | 0.9472 | **0.9709** | 0.8944 | 0.9659 |
| KS | | 0.2359 | 0.2916 | **0.2275** | **0.1983** | 0.2997 | 0.2434 | 0.5105 | **0.1318** | 0.2428 | **0.2032** | 0.3203 | 0.2798 |

obtained directly from the dataset itself and enable the design of a parametric version of Sammon's mapping that avoids out-of-sample extension, similar to the parametric extension of t-SNE proposed in (van der Maaten, 2009).

# 3 CHANNEL CHARTING WITH REPRESENTATION CONSTRAINTS

We now showcase an application example of representation constraints in wireless positioning. In particular, we augment the channel charting (CC) framework of Studer et al. (2018) for unsupervised user positioning with representation constraints that naturally arise from the data and application itself. We start by outlining the key concepts of CC and then explain how representation constraints can be included. We then demonstrate the efficacy of our approach in comparison to the original CC method.

## 3.1 CHANNEL CHARTING BASICS

CC measures CSI from users at different spatial locations and learns a low-dimensional *channel chart* that preserves *locally* the original spatial geometry. Put simply, users that are nearby in physical space will be placed nearby in the channel chart and vice versa—global geometry is not preserved. In this framework, high dimensional features are extracted from CSI, then processed with dimensionality-reduction methods to obtain the low-dimensional channel chart. CC operates in an unsupervised manner, i.e., learning is only based on CSI that is passively collected at an infrastructure base-station (and required anyway for data detection and precoding) but from multiple user locations in the service

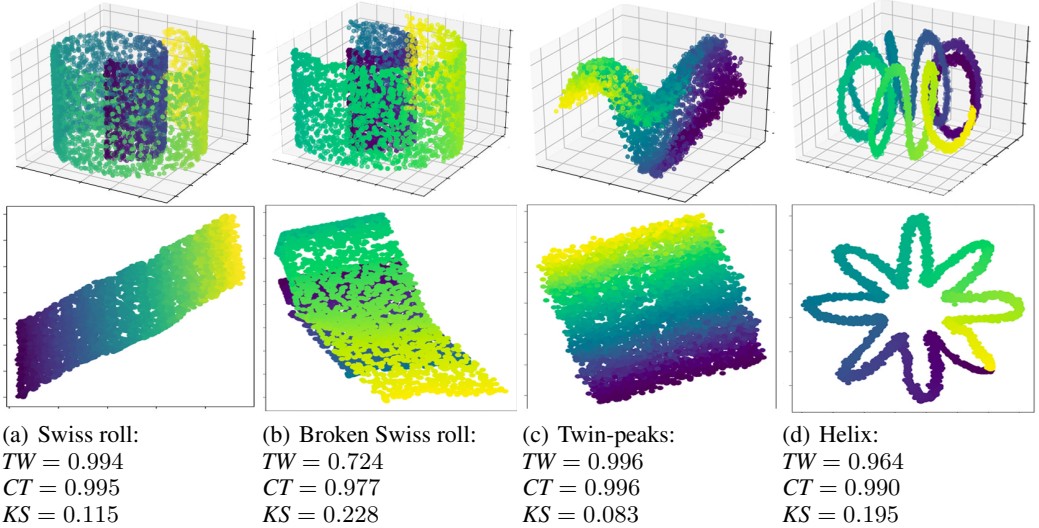

| (a) Swiss roll: | (b) Broken Swiss roll: | (c) Twin-peaks: | (d) Helix: |
|---|---|---|---|
| $TW = 0.994$ | $TW = 0.724$ | $TW = 0.996$ | $TW = 0.964$ |
| $CT = 0.995$ | $CT = 0.977$ | $CT = 0.996$ | $CT = 0.990$ |
| $KS = 0.115$ | $KS = 0.228$ | $KS = 0.083$ | $KS = 0.195$ |

Figure 1: Top row: synthetic datasets in three-dimensions; bottom row: two-dimensional embeddings obtained via AEs with FRD constraints to emulate Sammon's mapping. Except for the broken Swiss roll, AEs with FRD constraints almost perfectly reveal the low-dimensional manifold structure.

area over time. CC opens up many location-based applications as it provides base-station providers with relative user location information without access to GPS or fingerprinting methods (Liu et al., 2007) that require expensive measurement campaigns.

The technical concepts behind channel charting are as follows. Assume that we have $N$ single-antenna users located in real space with coordinates $\mathbf{z}_n \in \mathbb{R}^3$. If the $n$th user at location $\mathbf{z}_n$ is transmitting data to a base-station (BS), then the BS extracts first high-dimensional CSI in the form of a high-dimensional vector $\mathbf{h}_n \in \mathbb{C}^D$, which represents multi-path scattering and path loss of the wireless channel. From the CSI vector $\mathbf{h}_n$, one can extract features $\mathbf{x}_n \in \mathbb{R}^D$ that represent large-scale fading properties of the wireless channel. The main assumption of CC is that large-scale fading properties are mostly static and strongly depend on user location. Specifically, due to the underlying physics of electromagnetic wave propagation, each CSI feature is a (noisy) function of user position that represents the effect of the (unknown) physical environment on the wireless channel of the transmitted signal. One can then learn the channel chart from the set of channel features $\{\mathbf{x}_n\}_{n=1}^N$ in an unsupervised manner by means of dimensionality reduction methods. If AEs are used in this procedure, the encoder $f^e$ corresponds to the forward charting function that maps CSI features $\mathbf{x}_n$ to relative position information $\mathbf{y}_n$ in represenation space.

Studer et al. (2018) proposed the use of Sammon's mapping (SM) and AEs to learn the channel charts. While SM exhibited good performance, AEs scale well to large problem sizes and provide a parametric mapping that enables one to map new, unseen CSI features to a relative location information. Despite the advantages of AEs, valuable side information that arises from the application itself has been ignored. First, in contrast to SM, conventional AEs do not enforce any geometric structure on their representations. Second, by tracking a user's CSI over time, the corresponding low-dimensional representations that reflect their position should be similar as velocity is limited. Third, absolute positioning is currently not possible with the original CC framework.

## 3.2 Channel Charting with Representation Constraints

We propose the inclusion of representation constraints to overcome the limitations of the original CC framework. Concretely, we impose MAD constraints on pairs of representations from a user over time in order to ensure that nearby spatial locations for nearby representations. We can estimate an upper limit on the maximum distance in representation space dependent on the measurement CSI acquisition times. Note that this information comes from the CSI measurement process and the

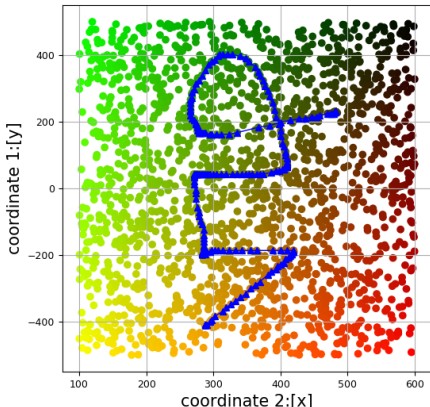

Figure 2: Positioning scenario modeled after (Studer et al., 2018). A 32-antenna base-station antenna array located at the origin extracts CSI from 2000 users in space. Each point represents a user in space; the points on the "vip" curve model a user that moves in time. The color gradients and "vip" curve enable an easy visual comparison with the learned channel charts shown in Figure 3.

fact that we know how data was collected in a real system. This approach remains passive as no supervision or measurement campaigns are necessary.

Furthermore, to enable true positioning capabilities with CC, we unwrap the channel chart using what we call *anchor vectors*, i.e., points in space for which we know both their CSI as well as their true location. One can imagine measuring CSI at a small set of locations when setting up a new base-station. With this information, we can impose FAD representation constraint on the AE with $\underline{d}_{i,j} = 0$ to enforce the exact anchor positions. We note that the inclusion of such constraints leads to a semi-supervised version of CC (and AEs in general) and requires measurement campaigns. We emphasize, however, in contrast to conventional fingerprinting methods that are fully supervised and require training at wavelength resolution in space, we only require a small number of anchor vectors and use the rest of the (unlabeled) data to improve the localization accuracy of the channel chart.

### 3.3 NUMERICAL RESULTS

We now provide numerical results for CC with representation constraints.

**Scenario** Figure 2 depicts the scenario provided in (Studer et al., 2018). We measure the CSI of $N = 2048$ randomly placed user locations (with the exception of the positions on the "vip" curve) within a rectangular area of $1000\,\text{m} \times 500\,\text{m}$. We model the acquisition of CSI at $0\,\text{dB}$ SNR. At spatial location $(x, y, z) = (0, 0, 10)$ meters, we consider a uniform linear BS antenna array with half-wavelength spacing and 32 antennas. We model data transmission at $2\,\text{GHz}$ and consider three channel models: (i) Vanilla line-of-sight (V-LoS) is a textbook channel model that enables a closed-form expression for the CSI (Tse & Viswanath, 2005); (ii) Quadriga LoS (Q-Los) is a realistic model for LoS channels that includes scatterers and path loss; and (iii) Quadriga non-LoS (Q-NLoS) is a realistic model for non-LoS channels (no direct path from users to BS antenna array). For both Quadriga models, we used the "Berlin LoS UMa scenario," which has been calibrated with real-world measurements (Jaeckel et al., 2014). At the base station side, we extract the same set of features as proposed in (Studer et al., 2018), i.e., we apply feature scaling, convert them into the angular domain, and take the entry-wise absolute value. The input thus has $D = 32$ real dimensions; the representation dimension is $D' = 2$. We use a similar structure of the AE as in (Studer et al., 2018), namely 9 hidden dense layers in total: 4 dense layers for the encoder part and 4 dense layers for the decoder part (the layers consist of 500, 100, 50 and 20 neurons), and an intermediate layer where we extract the 2-dimensional channel chart with 2 neurons and linear activation functions.

**Results** Figure 3 shows the channel charts. The top row (Figures 3(a)–(c)) shows the results of a "plain" AE, i.e., without any representation constraints; these results reproduce those in (Studer et al., 2018). The middle row (Figures 3(d)–(f)) shows channel charts from AEs that include FAD constraints, where we use a fraction of $10\%$ of randomly-selected anchor vectors. Clearly, these FAD

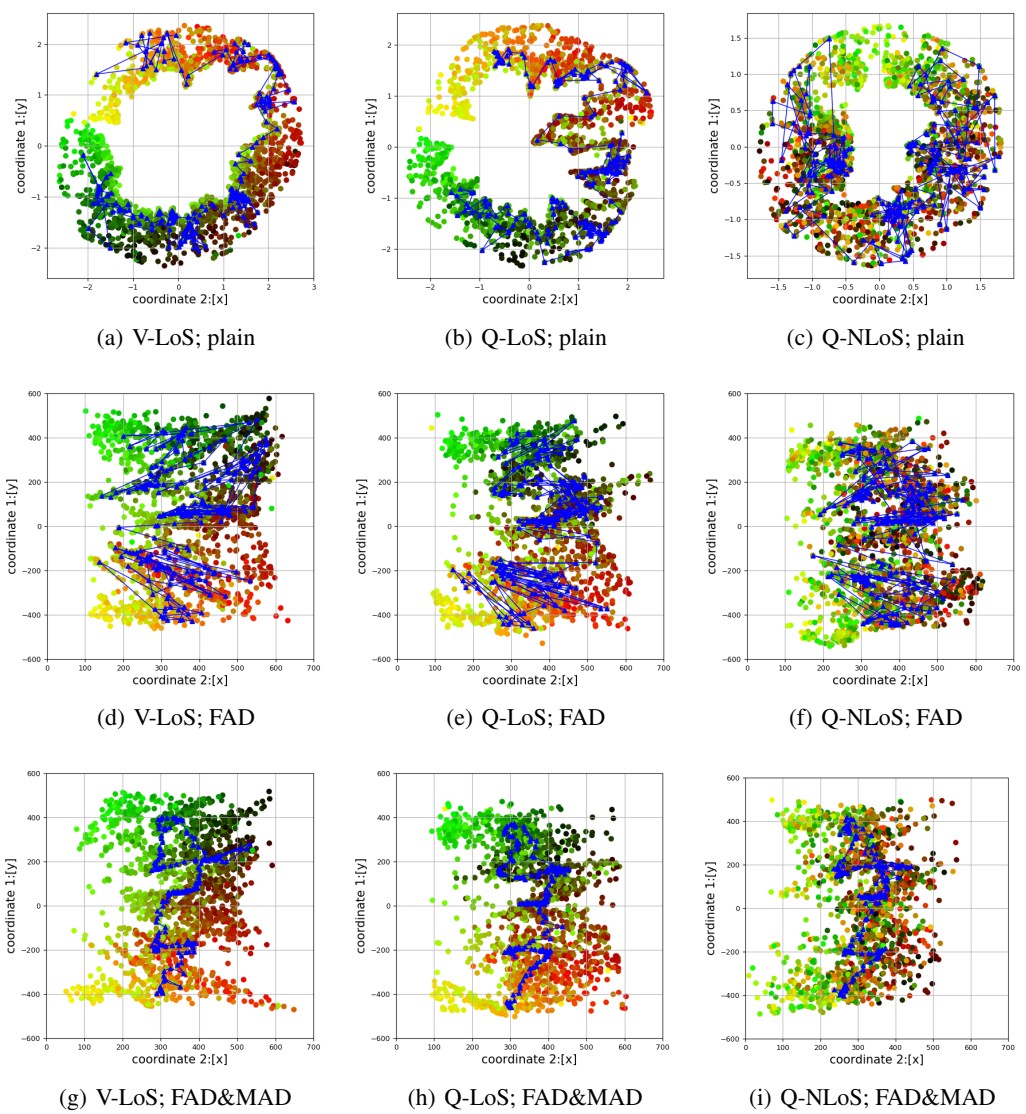

Figure 3: Channel charts learned from the scenario in Figure 2. The channel charts of plain AEs represent well the local neighborhood structure, but do not capture the global geometric properties. Imposing FAD representation constraints recovers parts of the global geometry. Imposing FAD and MAD constraints recovers the fine aspects of the geometry, enabling approximate positioning.

constraints unwrap the channel chart and lead to a representation with a distance scale comparable to the original scenario in Figure 2. While the global structure is approximately preserved, the points on the "vip" curve are not represented accurately. The bottom row (Figures 3(g)–(i)) shows the combination of FAD and MAD constraints in AEs. Concretely, we now also enforce the fact that points on the "vip" curve model a user's motion and we can impose maximum absolute distance constraints among pairs of representations pertaining to this curve. Clearly, FAD and MAD combined are able to reproduce the original scenario (Studer et al., 2018). While the representation is not perfect, it is evident that points in the channel chart approximately represent real locations. We also see that the propagation conditions of the wireless channel do not substantially affect the performance of the considered methods.

Table 3 shows the TW, CT, and KS results for the channel charts shown in Figure 3. We see that including representation constraints improves TW while slightly lowering the CT; note that TW and CT are evaluated for $K = 1$, $K = 51$ (2.5% of the dataset), and $K = 102$ (5% of the dataset).

Table 3: TW, CT, and KS results for channel charting with and without representation constraints.

|     |          | V-LoS | | | Q-LoS | | | Q-NLoS | | |
| --- | --- | --- | --- | --- | --- | --- | --- | --- | --- | --- |
|     |          | Plain | FAD | FAD&MAD | Plain | FAD | FAD&MAD | Plain | FAD | FAD&MAD |
| TW  | $K = 1$   | 0.8979 | 0.8860 | **0.8980** | 0.8468 | 0.8516 | **0.8576** | 0.8480 | 0.8492 | **0.8597** |
|     | $K = 51$  | 0.8974 | 0.8914 | **0.9079** | 0.8597 | 0.8570 | **0.8651** | 0.8502 | 0.8560 | **0.8665** |
|     | $K = 102$ | 0.8920 | 0.8967 | **0.9129** | 0.8609 | 0.8642 | **0.8736** | 0.8546 | 0.8626 | **0.8739** |
| CT  | $K = 1$   | **0.9655** | 0.9168 | 0.9436 | **0.9700** | 0.9195 | 0.9321 | **0.9281** | 0.8924 | 0.9110 |
|     | $K = 51$  | **0.9532** | 0.9060 | 0.9354 | **0.9440** | 0.9067 | 0.9215 | **0.9168** | 0.8928 | 0.9128 |
|     | $K = 102$ | **0.9480** | 0.9092 | 0.9375 | **0.9358** | 0.9081 | 0.9216 | **0.9155** | 0.8964 | 0.9151 |
| KS  |          | 0.3620 | 0.2514 | **0.2351** | 0.3548 | 0.2652 | **0.2598** | 0.4096 | 0.2749 | **0.2693** |

Hence, we observe a tradeoff with respect to neighborhood-preserving properties. More concretely, an increase in TW means that we are introducing less "fake" near neighbors; a reduction in CT means that original neighborhoods in the original space are not as well preserved in the channel chart as before. With respect to the global geometric structure, we see that KS significantly improves for all representation-constrained AEs; this implies that the inclusion of constraints enables us to recover global geometry. We note that this is also visible in Figure 3, especially in the AE results that include both FAD constraints (anchor vectors) and MAD constraints (to enforce continuity of a user's motion over time). Once again, we see that the propagation conditions do not substantially affect the performance of our CC.

## 4 CONCLUSIONS

We have proposed representation-constrained autoencoders (AEs) which enable us to impose structure on pairs of representations. We have shown, for a set of synthetic datasets, that only a small fraction of representation constraints already yields significant improvements in terms of trustworthiness, continuity, and Kruskal stress. To provide an application example for representation constraints in a realistic scenario, we have shown an improvement to the channel charting (CC) framework in (Studer et al., 2018), where we use side information on user motion and anchor vectors to improve the positioning performance of CC. Numerical results for this application have shown that the use of representation constraints that are readily available in wireless positioning scenarios yield (often significant) improvements in recovered global geometry.

There are many opportunities for future work. Parameter tuning of the regularizers required for each set of constraints is time consuming. The development of a systematic and automated way to tune the parameters of CC would significantly improve the efficacy of our framework. We have shown that our framework enables a parametric version of Sammon's mapping—a systematic investigation of this method is left for future work. While our methods enable approximate positioning, AEs with representation constraints are still not sufficient to enable GPS-grade positioning performance. Promising research directions towards this goal are the development of improved CSI features as well as the inclusion of additional geometric constraints, e.g., when acquiring CSI from multiple cell-towers or access points.

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
