# OpenReview forum: "Representation-Constrained Autoencoders and an Application to Wireless Positioning"
_ICLR.cc/2019/Conference_

### Official Review · AnonReviewer3 · 2018-11-04
**Learning representation-constrained autoencoders**

**Rating:** 6
**Confidence:** 2

**Review:**

The paper propose to learn autoencoders which incorporate pairwise constraints while learning the representation. Such constraints are motivated from the available side information in a given application. Inducing application-specific structure while training autoencoders allows to learn embeddings with better neighborhood preserving properties. In wireless positioning application, the paper proposes fixed absolute/relative distance and maximum absolute/relative distance constraints. Experiments on synthetic and real-word datasets show improved performance with the proposed approach.

Some comments/questions:

1. Table 1 shows different constraints along with the corresponding regularizers which are employed while training autoencoders. How is the regularization parameter set for (so many) regularizers?

2. Employing constraints (e.g. manifolds) while learning representation has recently attracted attention (see the references below). The proposed approach may benefit from learning the constraints directly on the manifolds (than via regularizers). Some of the constraints discussed in the paper can be modeled on manifolds.

Arjovsky et al (2016). Unitary evolution recurrent neural networks
Huang et al (2017). Orthogonal weight normalization: Solution to optimization over multiple dependent Stiefel manifolds in deep neural networks.
Huang et al (2018). Building deep networks on Grassmann manifolds
Ozay and Okatani (2018). Training CNNs with normalized kernels.

---

> ### Author Response · Authors · 2018-11-06
> **Thanks a lot for the comments!**
>
> 1. We are only using one type of regularizer at once (but a sum over many pairs of points). Hence, we only end up with one regularization parameter that must be tuned. To select the best parameter in practice, we use a simple grid search. In cases where one wants to use multiple different types of regularizers, selecting the best parameters requires a multidimensional grid search. We have not investigated more efficient (or even automated) ways to select the regularization parameter in practice.
>
> 2. Thanks for pointing out these references. Whether these manifold constraints would help the performance of our algorithms is indeed an interesting question as we have not experimented with such methods in the context of our paper. In our application, we would like to impose such constraints not on the weights but rather on the embedded points. For example, enforcing orthogonal rows on a batch of embedded points may help to learn a more meaningful low-dimensional representation, similar to PCA. We will include the suggested references and outline potential applications of such manifold constraints on either the weights or the embedded points (which is more challenging, but may be more relevant in our scenario).

---

### Official Review · AnonReviewer2 · 2018-11-07
**Useful approach, but insufficient experimental validation, and somewhat weak on novelty**

**Rating:** 4
**Confidence:** 4

**Review:**

Description:

This paper presents a variant of deep neural network autoencoders for low-dimensional embedding, where pairwise constraints are incorporated, and applies it to wireless positioning.

The four constraint types are about enforcing pairwise distance between low-dimensional points to be close to a desired value or below a maximal desired value, either as an "absolute" constraint where one point is fixed or a "relative" constraint where both points are optimized. The constraints are encoded as  nonconvex regularization terms. In addition to the constraints the method has a standard autoencoder cost function.

Authors point out that if a suitable importance weighting is done, one constraint type yields a parametric version of Sammon’s mapping.

The method is tested on four simple artificial manifolds and on a wireless positioning task.


Evaluation:

Combining autoencoders with suitable additional regularizers can be a meaningful approach. However, I find the evaluation of the proposed method very insufficient: there are no comparisons to any other dimensionality reduction methods. For example, Sammon's mapping is mentioned several times but is not compared to, and a parametric version of t-SNE is also mentioned but not compared to even though it is parametric like the authors' proposed method. I consider that to be a severe problem in a situation where numerous such methods have been proposed previously and would be applicable to the data used here.

In terms of novelty I find the method somewhat lacking: essentially it is close to simply a weighted combination of an AE cost function and a Sammon's mapping cost function when using the FRD constraints. The other types of constraints add some more novelty, however.



Detailed comments:


"Autoencoders have been shown to consistently outperform other dimensionality-reduction algorithms on real-world datasets (van der Maaten et al., 2009)": this is too old a reference, nine years old, and it does not contain numerous dimensionality reduction algorithms proposed more recently, such as any neighbor embedding based dimensionality reduction methods. Moreover, the test in van der Maate et al. 2009 was only on five data sets and in terms of a continuity measure only, too little evidence to claim consistent outperforming of other algorithms.

"van der Maaten (2009) proposes the use of AEs to learn a parametric mapping between high-dimensional datapoints and low-dimensional representations by enforcing structure obtained via Student-t stochastic neighborhood embedding (t-SNE)": this is not a correct description, van der Maaten (2009) optimizes the AE using t-SNE cost function (instead of running some separate t-SNE step to yield structural constraints as the description seems to say).

"the FRD regularizer resembles that of Sammon's mapping": actually in the general form it resembles the multidimensional scaling stress; it only becomes close to Sammon's mapping if you additionally weight each constraint by the inverse of the original distance as you suggest.

It is unclear to me where the absolute distance constraints (FAD or MAD) arise from in the synthetic experiments. You write "for FAD one of the two representations... is a constant known prior to AE learning": how can you know the desired low-dimensional output coordinate (or distance from such a coordinate) in the synthetic data case?

This reference is incorrect: "Laurens van der Maaten, Eric Postma, and Jaap Van den Herik. Dimensionality reduction: A comparative review. In Journal of Machine Learning Research, volume 10, pp. 66–71, 2009." This article has not been published in Journal of Machine Learning Research. It is only available as a technical report of Tilburg University.

---

> ### Author Response · Authors · 2018-11-08
> **Thanks for the detailed set of comments**
>
> In what follows, C: stands for the reviewer's comment and R: for our response.
>
> C: "[...] I find the evaluation of the proposed method very insufficient: there are no comparisons to any other dimensionality reduction methods. [...]"
>
> R: We emphasize that our paper is not on dimensionality reduction per se, but rather on the inclusion of  side constraints and their use in the application for wireless positioning. We do not propose new methods that might outperform other methods (such as t-SNE, Sammon’s mapping, isomap, neighbor embedding, etc.) at conventional dimensionality reduction tasks. Instead, we are interested in enforcing side information on the low-dimensional representation which may be available in certain applications. We have decided to focus on augmenting autoencoders with side constraints (rather than parametric t-SNE or others) for the following reasons: (i) they scale favorably to a large number of datapoints together with stochastic gradient descent, (ii) they provide a parametric mapping and the learned neural net can be used in real-time applications, (iii) AEs have already worked reasonably well for channel charting without side information (so a direct comparison with the state-of-the-art for channel charting was possible), and (iv) existing deep learning frameworks enable an efficient development of autoencoders in practice. We admit that other methods, such as the parametric version of t-SNE, may work as well or even better in our application, but our goal was not to perform a comparison of methods that *could* work---we were rather interested in developing one method that addresses the shortcomings of channel charting. The evaluation of potentially better methods is left for future work.
>
> We admit, however, that for the synthetic experiments, we should have included a comparison to conventional Sammon mapping and t-SNE. We will do that in the final version.
>
> C: "In terms of novelty I find the method somewhat lacking: [...]. The other types of constraints add some more novelty, however.”
>
> R: The main focus of the paper was not on the Sammon’s mapping cost function but on including side constraints to autoencoders to enable positioning via channel charting. We see the fact that the FRD constraints enable a parametric version of Sammon’s mapping as a byproduct of our approach. Hence, we included the synthetic simulation results to showcase that our constraints are not only useful for wireless positioning.
>
> C: "Autoencoders have been shown to consistently outperform other dimensionality reduction algorithms on real-world datasets (van der Maaten et al., 2009)": this is too old a reference [...]"
>
> R: We used this reference as part to motivate the use of autoencoders and not other methods for our purposes. Furthermore, in the original channel charting paper, AEs were achieving better performance than other dimensionality reduction methods. Again, we are not proposing a new and better dimensionality reduction method for general tasks, but rather augmented an existing method (autoencoders) with our side constraints to improve applications that benefit from side information, such as channel charting. We will adjust the motivation in the final version of the paper..
>
> C: "van der Maaten (2009) proposes the use of AEs to learn a parametric mapping between high-dimensional datapoints and low-dimensional representations by enforcing structure obtained via Student-t stochastic neighborhood embedding (t-SNE)": this is not a correct description, [...]”
>
> R: By “enforcing structure obtained via Student-t stochastic neighborhood embedding” we meant that we learn the AE with the t-SNE cost function. We will rephrase our statement to more accurately reflect what is happening.
>
> C: "the FRD regularizer resembles that of Sammon's mapping": actually in the general form it resembles the multidimensional scaling stress; [...]
>
> R: Indeed, the plain FRD regularizers implements MDS. We will add this insight to the final version.
>
> C: "It is unclear to me where the absolute distance constraints (FAD or MAD) arise from in the synthetic experiments. [...]"
>
> R: Our synthetic experiments demonstrate that only a small amount of side information is sufficient to (often significantly) improve autoencoders. To showcase this behavior on synthetic data, we have used knowledge available from the true low-dimensional point set (used to generate the high-dimensional dataset). For example, for the fixed absolute distance (FAD) constraint we extract the distance between points in the low-dimensional representation and use a small fraction of  information while training the autoencoder. Clearly, this would not be possible in many dimensionality reduction applications. However, in our wireless positioning application, we can get the maximum distance between two points from the physics: a mobile terminal can only move at finite velocity which implies that subsequent CSI measurements cannot be too far apart in representation space.

---

### Official Review · AnonReviewer1 · 2018-11-12
**Applies a distance constraint to the latent space of auto-encoders**

**Rating:** 5
**Confidence:** 4

**Review:**

[I'm a fallback reviewer assigned after initial reviewer failed to submit]

Quality/Clarity:
The work is fine. The presentation is clear enough. The experiments are all on simulated data, with 2GHz scattering simulation derived from more sophisticated software suites than the 4 toy manifold problems initially considered.


Originality/Significance:
The work does not seem particularly novel. Perhaps the specific application of regularized autoencoders to the channel charting problem is novel. The regularizers end up looking a lot like a variety of margin losses. The idea of imposing some structure on the latent space of an autoencoder is not particularly new either. Consider, for example, conditional VAEs. Or this work from last year's ICLR https://openreview.net/forum?id=Sy8XvGb0- This work is straightforward multi-task learning with dimensionality reduction with similarity loss tasks.

On the whole, I don't think there is enough novel work for the venue.

---

> ### Author Response · Authors · 2018-11-14
> **Differences to conditional VAEs and related approaches**
>
> Thank you for your comment.
>
> We agree that the general idea of constrained representations in autoencoders is not novel per se. However, the type of constraints we are imposing are novel and the application to wireless positioning is new as well.
>
> In the references you mentioned, constraints are added for the purpose of conditionally generating data. The constraints are used to (i) learn a latent variable that would offer a better output reconstruction, (ii) allow for a realistic (and diverse) data generation, or (iii) enable more control over the characteristics of the generated data (e.g., controlling attributes). For example, conditional GANs (CGAN) and conditional VAEs (CVAE) introduce conditioning using extra information (e.g., attribute labels) during training phase to influence the distribution in the latent space with the goal of learning better structure in the outputs. In our case, we are not trying to train a generative model but rather perform dimensionality reduction to learn a “meaningful” low dimensional embedding. Since we care about the geometry of the latent variable rather than the reconstructed output of the AE, we impose side-information constraints on pairs of data points in the low-dimensional representation (i.e, latent variable). We do not put a specific emphasis on the probabilistic inference between the latent variable and the output of the autoencoder, and thus on the distribution of the latent variable. Also, our constraints (pairwise distances) are not of probabilistic nature. This is because the primary goal is to find a low dimensional representation that preserve a (hidden) geometrical structure of the input data (e.g., finding relative positions from channel state information features in wireless positioning).
>
> Our results are useful for applications where side information about the low dimensional representation arise naturally from the data or the application. For example, for positioning we wish to learn a 2D representation from high-dimensional channel state information features. Knowing (from the original channel charting paper), that autoencoders can find better embeddings (in terms of preserving the neighborhood) than other dimensionality reduction methods, we show that adding side information that arise from the user equipments’ finite velocity, or from some labeled positions (in a semi-supervised fashion), can help to significantly improve positioning.
>
> We will include the suggested references on latent-variable constrained generative models as a reference to representation constrained models and highlight their differences to our work.

---

> > ### Comment · AnonReviewer1 · 2018-11-26
> > **Replies to comments**
> >
> > I don't find the distinction with CVAE very convincing. I'd grant that this is not a variational model. But it's stretching to suggest autoencoders are not generative models (they are simply deterministic ones). The probabilistic interpretation is that we have a prior of the latents being 'nearby' to other latents from nearby in time. While there may not be particular emphasis on the probabilistic interpretation for each penalty metric, using a squared distance specifically suggests a Gaussian latent space of representations.

---

### Meta-Review · Area_Chair1 · 2018-12-13
**Interesting application to wireless positioning, but lacks novelty and empirical comparisons**

**Confidence:** 5
**Recommendation:** Reject

**Metareview:**

The reviewers found the work interesting and sensible.  The application of latent space constrained autoencoders to wireless positioning certainly seems novel.  Applications can certainly be exciting additions to the conference program.  However, the reviewers weren't convinced that the technical content of the paper was sufficiently novel to be interesting to the ICLR community.  In particular, the reviewers seem concerned that there are no comparisons to more recent methods for dimensionality reduction and learning latent embeddings, such as variational auto-encoders.  Certainly a comparison to more recent work constraining latent representations seems warranted to justify this particular approach.